# Contribution of Live Video to Physicians’ Remote Assessment of Suspected COVID-19 Patients in an Emergency Medical Communication Centre: A Retrospective Study and Web-Based Survey

**DOI:** 10.3390/ijerph20043307

**Published:** 2023-02-14

**Authors:** Robert Larribau, Beth Healey, Victor Nathan Chappuis, Dominique Boussard, Florent Guiche, Tara Herren, Birgit Andrea Gartner, Laurent Suppan

**Affiliations:** Division of Emergency Medicine, Department of Anaesthesiology, Clinical Pharmacology, Intensive Care and Emergency Medicine, Geneva University Hospital, Rue Gabrielle-Perret-Gentil 4, CH 1211 Geneva, Switzerland

**Keywords:** dispatch, emergency medical dispatch, emergency medical communication centre, video live, COVID-19, emergency call, video triage, public safety answering point, telemedecine, emergency medical services, remote assessment, triage

## Abstract

The COVID-19 pandemic had a major impact on emergency medical communication centres (EMCC). A live video facility was made available to second-line physicians in an EMCC with a first-line paramedic to receive emergency calls. The objective of this study was to measure the contribution of live video to remote medical triage. The single-centre retrospective study included all telephone assessments of patients with suspected COVID-19 symptoms from 01.04.2020 to 30.04.2021 in Geneva, Switzerland. The organisation of the EMCC and the characteristics of patients who called the two emergency lines (official emergency number and COVID-19 number) with suspected COVID-19 symptoms were described. A prospective web-based survey of physicians was conducted during the same period to measure the indications, limitations and impact of live video on their decisions. A total of 8957 patients were included, and 2157 (48.0%) of the 4493 patients assessed on the official emergency number had dyspnoea, 4045 (90.6%) of 4464 patients assessed on the COVID-19 number had flu-like symptoms and 1798 (20.1%) patients were reassessed remotely by a physician, including 405 (22.5%) with live video, successfully in 315 (77.8%) attempts. The web-based survey (107 forms) showed that physicians used live video to assess mainly the breathing (81.3%) and general condition (78.5%) of patients. They felt that their decision was modified in 75.7% (n = 81) of cases and caught 7 (7.7%) patients in a life-threatening emergency. Medical triage decisions for suspected COVID-19 patients are strongly influenced by the use of live video.

## 1. Introduction

The Coronavirus (COVID-19) pandemic had major implications for emergency medical communication centres (EMCC) [1,2,3], which had to be reorganised to respond effectively to the large influx of telephone calls from patients presenting with influenza-like illness and dyspnoea [4,5,6]. Although online triage systems were developed [7,8], the increase in calls to EMCCs due to the large number of respiratory distress situations was very significant [9]. Some EMCCs developed a two-tiered response, where the first tier performed the initial triage (differentiating between sick/not-sick and mild/severe symptoms) and the second tier was comprised of clinicians (emergency physicians, advanced practice nurses or advanced paramedics) who assessed more complex and severe cases [1,5]. The aim of the EMCC restructuring was to improve the efficiency of telephone triage and avoid, where possible, large uncontrolled influxes of patients attending medical facilities [10,11].

Since the widespread utilization of smartphones with video capability and concurrently improving satellite connectivity, live video telemedicine calls have presented exciting possibilities for improving patient care. Before the pandemic, the use of live video in EMCCs was in its infancy [12,13]. However, some centres had already tried to replace the dispatch of emergency physicians (EP) to support paramedics in the field with remote assistance [14,15]. A study conducted just prior to the pandemic demonstrated that the addition of live video to an emergency call was feasible and changed the perception of the emergency medical dispatcher in over half of cases [16]. Another study showed that live video guidance (by Emergency Medical Dispatchers) of resuscitation manoeuvres resulted in a subjective improvement in the quality of resuscitation [17].

The COVID-19 pandemic was a catalyst in the use of live video as this capability limited the necessity of physician-patient contact and helped meet the increased demands placed on services during this period [18]. The use of live video during the pandemic subsequently became increasingly prevalent and helped improve quality of care, in particular in the detection of respiratory distress and identifying the requirement for emergency care [19,20].

To date, very few studies have been published which consider the reorganisation of EMCCs during the pandemic. No study has specifically measured the activity of physicians who triaged second-line emergency calls in EMCCs during the COVID-19 pandemic. There has also been no study to date measuring the use and impact of live video utilisation by second-line physicians in an EMCC.

The main objective of this study was therefore to measure the effect of the use of live video on medical decisions made by EMCC physicians in the remote assessment of suspected COVID-19 patients during the first two waves of the pandemic.

Secondary objectives were to measure the frequency, relative indications and limitations of the use of live video in an EMCC, to describe the characteristics of patients assessed by helpline staff and the reorganisation of an EMCC in response to the pandemic.

## 2. Materials and Methods

Both the observational study and web-based survey were conducted during the COVID-19 pandemic in Geneva (Switzerland) from 1 April 2020 until 30 April 2021. Data was derived from (1) the call assessment registry of Geneva Emergency Medical Communication Centre (EMCC) and (2) a closed web-based survey of EMCC physicians which was conducted at the same centre. The studies followed the STROBE [21] and CHERRIES guidelines, respectively [22].

### 2.1. Setting

The Geneva canton covers an area of 282.48 km^2^, is predominantly urban and had a recorded population of 508,774 at the outset of the study [23]. Twenty one percent of residents were under 20, 16% over 64 and 52% women. There were additionally 100,000 cross-border workers commuting daily from France or neighbouring Swiss cantons to work in Geneva during this period [23].

### 2.2. Geneva’s Typical EMCC and Emergency Medical System (EMS)

Geneva’s EMCC receives all health-related emergency calls in the canton, and the EMDs (paramedics or registered nurses) carried out approximately 40,000 telephone assessments of patients in 2019 (excluding inter-hospital transfers). The EMDs are either registered nurses or paramedics with a minimum of 5 years of field experience. They handle all calls using a computer-aided dispatch system (ICAD^®^ from Hexagon AB^®^). For telephone assessments of patients, Geneva’s EMDs have used the symptom-based dispatch (SBD) system derived from the Swiss Emergency Triage Scale (SETS^®^) since 2013 [24]. Historically, there was not an Emergency Physician (EP) physically present in the EMCC, although EMDs had always had remote access to a physician in case of an assessment concern or medical question.

In the canton of Geneva, the EMS is two-tiered (or three) with different medical levels and skill sets. The first level is made up of ambulances, staffed by paramedics. There are fifteen ambulance bases scattered throughout the canton of Geneva that operate according to the proximity of the base to the emergency site. The second level consists of a Mobile Emergency and Resuscitation Service or SMUR, i.e., a light vehicle that operates with a certified paramedic and an emergency physician in training with at least 2 years of experience [20]. To assist this junior emergency physicians or if the SMUR is already busy with another emergency event, specialised senior emergency physicians are available 24 h a day, 7 days a week to intervene on the spot (third level). The senior EP is available 24/7 to support both levels of response remotely and in the field.

For non-vital cases EMDs also have the option of contacting one of the 3 on-call general practitioner (GP) services, to arrange a home visit. In practice, there are typically between 1 to 5 GPs available, often with no night cover. In view of the limitations of this service it is relatively infrequently used.

### 2.3. Re-Organization of Geneva’s EMCC during the COVID-19 Pandemic

At the outset of the COVID-19 pandemic (6 March 2020), Geneva’s EMCC was reorganized. Calls were separated into two streams: those made on the usual ‘144′ emergency number and those related to the COVID-19 pandemic, made on a new dedicated COVID-19 helpline. The staff responding to this new COVID-19 helpline were medical students (between their 2nd and 4th years of study) and health professionals from a variety of backgrounds (dispatch assistants). They were specifically trained to assess patients with symptoms related to COVID-19, using simplified assessment protocols. The EMD staff responding to the ‘144′ emergency line remained unchanged, and the initial protocol for assessment of suspected COVID-19 patients was identical to that used by the other COVID-19 helpline (investigation of the main symptom presented by the patient, assessment of general condition and breathing).

In addition, a physician was made available to reassess difficult situations (EMCC Physician). Typically, they did not take first-line calls and were only involved in the assessment of patients with suspected COVID-19 infections. They came from a variety of medical specialties whose activities had been suspended due to the pandemic and rotated through the EMCC for between one and six months. They were physically present 24/7 for the first two months, but as the pandemic progressed, this was reduced to day and evening cover and thereafter daytime only, a schedule which was adapted in accordance with COVID-19 pandemic waves. The two main waves of infections linked to the COVID-19 pandemic affected the canton of Geneva in March–April 2020 and from mid-October to mid-December 2020. In 2021, two further increases in infections were observed between April and September 2021 [25].

Following assessment, there were three possible responses for the patient: (1) telephone advice, (2) home visit by a general practitioner (limited availability) or (3) EMS dispatch. In cases where patients remained at home following assessment by a general practitioner or ambulance team, there was also the option for a (<72 h) telephone follow-up by the EMCC physician. The call source, organization of call handling and response are described in Figure 1.

In addition, during the two waves of the pandemic, the organisation of hospitals in the canton of Geneva was reviewed, with the main public hospital (Geneva University Hospitals), which was defined as the “COVID-19 hospital”, receiving all patients with (or suspected of having) COVID-19, and the other hospitals or private clinics mainly receiving patients without COVID-19 [26].

### 2.4. Live Video Facilities

The Instantview^®^ live video application from the company Urgentime^®^ was also made available to EMCC physicians from the 20 March 2020. The live video system was simple to operate. The physician sent a web link via short message service (SMS), which, when accepted by the caller, allowed a web-based application to open a live video feed between a callers’ smartphone and the EMCC physician. This stream was unidirectional between the caller and physician (the physician sees the caller, but the caller does not see the physician). This video stream was in addition to and remained separate from the original audio stream.

The live video was made available to all EMCC physicians. Physicians were given the option of using live video when they felt it would add value to their remote assessment of suspected COVID-19 patients. There was no binding assessment protocol for the use of live video. The live video tool was only made available to them, and they had all been trained individually in its use. The EMCC physicians therefore used live video at their discretion for the assessment of suspected COVID-19 patients.

### 2.5. Web-Based Survey

An online survey form was created using the company Reallience^®^’s form creation tool available from the digital intelligence platform LogIC^®^ (https://www.logic-app.ch/, accessed on 1 April 2020). The people who could access the form were only those with a LogIC login and profile. A profile had been created for all staff working in the EMCC, but access was only given to physicians working in the EMCC. Instructions for use were provided orally and e-mailed to the physicians at the beginning of their rotation (on average one to three months).

The web-based form was a single page, structured questionnaire with a total of 16 questions. The first five questions identified the situation being assessed (Appendix A, questions 1–5). Next were 10 closed questions for which only one answer was possible (Appendix A, questions 6–15), and the last question was an open question (Appendix A, question 16). There were no conditional questions. The form could only be validated if all the questions were answered. The event number was requested to avoid duplication and was destroyed after the forms were checked and extracted.

Physicians were informed of the study’s purpose and of its estimated length. Physicians who filled in the form were under no obligation to do so (even if they were encouraged to do so), and so they freely agreed to answer and thereby allow the data to be used and published for scientific research purposes.

Both the internet platform and the questionnaire were thoroughly tested for usability and user-friendliness by several study authors before beginning the study. Identity and contact of the investigators were given, and information regarding data handling was provided.

### 2.6. Study Design

#### 2.6.1. Data

The data used in this retrospective study was derived from two sources: (1) register of telephone assessments and (2) paper documentation of daily activity recorded by EMCC physicians (which was used to measure the monthly volume of medical activity).

Survey data was obtained via the web-based form made available to all EMCC physicians (LogIC^®^), and live video use data was derived directly from the Instantview^®^ application.

#### 2.6.2. Inclusion and Exclusion Criteria

For the retrospective study, patient assessments made between 1 April 2020 and 30 April 2021 (13 months) were included. This represented the first two COVID-19 pandemic waves in Geneva. All calls received regarding a patient with a suspected or confirmed case of COVID-19 on both the 144 emergency and dedicated COVID-19 helplines were considered (“context” COVID-19). Assessments of patients already in hospital (inter-hospital transfers) were excluded. All data from the Instantview^®^ application regarding video calls made during this period were included.

For the web-based survey, all complete forms submitted were analysed.

#### 2.6.3. Outcomes

The primary outcome was the contribution of live video to medical decision making during remote assessment of suspected COVID-19 patients.

Secondary outcomes included the frequencies and characteristics of the populations assessed on the two emergency helplines, decisions made according to the lines called and the rate of assessments performed by a physician at Geneva’s EMCC. Secondary outcomes also included the frequency of physician live video use, reasons for using live video and limitations of using the platform.

#### 2.6.4. Measures

This study evaluated the total number of calls to the EMCC regarding patients with a suspected or confirmed case of COVID-19. This included all calls made to both the emergency (144) and dedicated COVID-19 lines. The activity of EMCC physicians and the rate of their live video use was also measured.

The data from the Urgentime^®^ live video application were analysed and the monthly video connection failure rate was additionally measured.

The results of the online survey, which focused on potential difficulties encountered and the estimated impact of live video on decision making, were also considered.

#### 2.6.5. Statistical Analysis

Comma-separated value (CSV) files containing data related to suspected COVID-19 patients from the Geneva SBD system registry and the web-based survey were imported into STATA^®^ 16.0 software (StataCorp^®^, College Station, TX, USA). Additional data (from the application Instantview^®^ and the count of paper forms) were processed directly in Microsoft Office^®^ Excel^®^ 2013.

Descriptive statistics calculations, including 95% confidence intervals, were performed using STATA^®^ 16.0 software. Student’s t-test was used for comparisons of means and the Chi square test (STATA 16.0) was used to compare proportions of categorical variables.

A test result was considered significant when *p* < 0.05.

## 3. Results

From 1 April 2020 to 30 April 2021, 53,566 telephone assessments were made at Geneva EMCC on both lines (emergency line and COVID-19 helpline). Of these, 8957 (16.7%) had a suspected or confirmed case of COVID-19. A total of 4464 (49.8%) assessments were made by an EMD via the emergency ‘144′ line whilst 4493 (50.2%) were made by medical students or dispatch assistants on the dedicated COVID-19 line.

During this period, 1798 (20.1%) of all COVID-19 assessments were provided to the EMCC Physician for further (second-line) evaluation. In 405 (22.5%) of these cases, EMCC physicians tried live video, of which 315 (77.8%) were considered successful and 90 (22.2%) unsuccessful. The flow chart below (Figure 2) describes the overall emergency assessment flow. Appendix A describes the monthly flow of remote assessments by physicians.

Significant differences were demonstrated between patients calling the 144 emergency line compared to the dedicated COVID-19 helpline (Table 1). Of these patients, it was found that the level of medical dispatch priority was significantly more urgent for patients calling the 144 emergency line (*p* < 0.001). In turn, significantly more ambulances were dispatched for patients calling the 144 emergency line (*p* < 0.001). The more severe levels of medical dispatch priority decided upon when calls were made to the 144 emergency line were later reflected in higher (more severe) NACA scores seen in the field than when calls were made to the COVID helpline19 (*p* < 0.001).

Of the 405 video assessments, the failure rate was 22.2% (n = 90) (Table 2). This data comes from the Instantview^®^ application database. Failure to use live video was either related to the users in the field (patients or callers) or to technical difficulties. Regarding user-related failures (slightly over half), not selecting the URL link in the SMS message comes first, followed by not allowing access to the smartphone camera. Regarding technical difficulties, the lack of a camera on the smartphone came first, followed by poor data connection.

The survey based on a web-based questionnaire (Table 3) demonstrated that patients evaluated by video were rather young and, in majority, female. Of the 107 (26.5%) evaluations with live video that were documented in the web survey forms by 13 EMCC physicians, acceptance was rated as “very good or excellent” for the majority of patients/callers, and ease of use was rated as “very good or excellent” by the majority of physicians. Most of the time, these patients were accompanied and the indication for assessment with live video was mainly to assess the patient’s breathing and general condition. For the majority of assessments, there were multiple indications for assessment using live video.

In Table 4, it was noted that after the live video assessment, 15 ambulances (16.5%) were dispatched (7 in Dispatch Priority Level 1) in 91 situations, where it had not been deemed necessary prior to live video assessment. In addition, it was found by physicians at Geneva EMCC that live video contributed to the outcome of their clinical decision in over 3/4 situations.

## 4. Discussion

Physicians responding to our web-based survey felt that live video contributed to a change in their second-line assessment of suspected COVID-19 patients in more than 75% of the situations. This rate is higher than in the Linderoth study, where the dispatchers’ assessment was changed in “only” 51.1% of cases [16]. This could be explained by a patient pre-screening bias, as the system is usually only used by physicians when a clear probable benefit has been pre-identified. Indeed, physicians used video in only 22.5% of their assessments when they could use it without any limitation. However, there was a similar rate (16.5% in our survey/12.9% in Linderoth’s study) of requalification in more critical situations or those requiring an ambulance. In the second-line live video assessments, there were a significant number of potentially life-threatening emergencies (7/91) that were detected (7.7%). The value of using live video, even after an advanced telephone assessment by a second-line physician, thus seems to persist. Another study showed that 22.4% of patients who were initially classified in the least severe category after telephone-only triage were reclassified to the most severe categories after reassessment using live video [27].

It would also appear that live video, used by a physician, could help to provide the “right answer” for the patient, even when paramedics are on site, since a redirection of the ambulance to the hospital and inpatient units dedicated to COVID-19 patients was performed in 10/16 situations. However, this study does not prove an advantage of live video, as there was no control group for this subgroup. Furthermore, another study concluded in the COVID-19 context that the use of live video by emergency physicians was not superior to the use of the telephone alone [28]. Nevertheless, it seems that nurses/paramedics have difficulty in making an adequate “no transport” decision when they are sent to the patient [29]. In these situations, having a remote physician could improve the appropriateness of transport decisions versus leaving the patient on site.

We noted that the failure rate of live video was 22.2%, which was slightly higher than the Danish study (17.8%) [17] and the English study (16.9%) [27]. It is possible that this is partly related to the fact that our EMCC only started using the Instantview^®^ tool on 20 March 2020, so there was only a very short learning phase for physicians. As in the Danish study, the causes of failure or difficulties were mainly related to the users’ handling of the smartphones or non-receipt/difficulty in accessing the SMS URL link. As with all published studies on the subject [17,27,28,30], acceptance of the video was excellent by both callers and physicians and was also considered easy to use.

We found significant differences between patients who called the 144 number and those who called the COVID-19 medical helpline. In particular, patients who called the COVID-19 helpline or those assessed via live video had a significantly lower mean age than those assessed after a call to the 144 number. This is comparable to studies that have assessed patients with suspected COVID-19, which found similar mean ages for both video and non-video assessments [5,27].

The predominant symptoms during the telephone assessment also differed according to the telephone lines that were called. Dyspnoea predominates for calls to the 144 number and flu-like symptoms for the COVID-19 medical helpline. In our EMCC, a fever was only mentioned if it was measured by the patient (the “feeling of fever” was not documented as a “fever”). This may explain the low proportion of fever documented in telephone assessments compared to other published studies [5,31]. Similarly, “dry cough” was not differentiated from other influenza symptoms. Taking these details into account, the majority of influenza-like symptoms were found in the mild forms of suspected COVID-19. Our study seems to show that, for suspected COVID-19, the presence of “dyspnoea” (including tachypnoea) at the initial telephone assessment does seem to be a criterion of severity at the time of the call.

The identification of dyspnoea or tachypnoea as a severity factor in remote assessments of suspected COVID-19 patients probably explains why, in our study, the assessment of breathing is the first cause of indication for the addition of live video that is mentioned by second-line physicians. Indeed, the measurement of respiratory rate, the search for accessory muscle use or cyanosis, are clearly facilitated by visualisation of the patient. This has been demonstrated in several studies, especially in the paediatric context [20,32,33].

The second main indication for the use of video was the “general condition of the patient”, which is often associated with the patient’s living situation in pre-hospital interventions. Live video allows the scene to be seen and can therefore provide valuable insights that can be missed completely by using the telephone alone. In the accident context, it has been argued that live video provides “more information from the scene of an incident and the clinical condition of the patient(s)” [13].

In our study, citizens could call two numbers, the usual emergency number in case an ambulance (number 144) was needed and another number dedicated to “remote medical assessments in case of suspected COVID-19” for situations that the callers themselves considered as “non-urgent”. Then, patient assessments were carried out in the same way, in the same EMCC, regardless of the telephone line initially called. We found that callers assessed their situation fairly well themselves, as only 5 (0.1%) serious situations (for 4464 assessments) were identified following a call to the COVID-19 medical helpline, compared with 735 (16.3%) serious situations (for 4493 assessments) on the 144 number (Table 1). The provision of two telephone helplines (urgent or ambulance/non-urgent) therefore contributes to the performance of the EMCC triage, as the prevalence of severity is completely different depending on which line is called [24]. The skills available and the assessment tools used in EMCCs should be adapted to the condition of the patients assessed. Medical skills and tools for live video are needed for patients requiring an ambulance, but also for low acuity situations [16,17,27,30,34].

The most likely outcome of a live video call in this study was for the patient to remain at home with no further follow-up required. It can therefore be inferred that the EMCC physician was able to completely satisfy a patient’s needs and no further resources were required in these cases. This can be considered a particularly successful outcome in view of the fact that the patients being reviewed by the physician were second line and therefore likely to be more complex scenarios. It was also particularly relevant to the COVID-19 pandemic where limiting patient-clinician contact reduced the likelihood of disease transmission. This outcome not only saves precious resources but is also positive for CO_2_ emissions, reducing the requirement of unnecessary journeys by prehospital teams or patients.

The exponential increase in calls to EMCCs during the pandemic, mainly for remote triage of patients suspected of COVID-19 [4], has resulted in a change in the missions of these EMCCs from “situation-related emergency medical dispatch” to “remote triage of patients”. In this context, EMCCs are increasingly performing “tele triage”, and this is part of the “virtually perfect” evolution of emergency telemedicine [35]. Finally, this advanced remote triage using telemedicine tools may have had a strong impact on avoiding hospital overcrowding during the pandemic and is therefore one of the tools to be developed for future crises [36,37].

There are several limitations to our study. Firstly, this was a retrospective, single-centre study. Few organisations are comparable to the EMCC in Geneva with two levels (first level with nurses or paramedics and second level with a physician), and this limits the generalisation of the results. Secondly, the second part was a web-based survey with all the known limitations associated with this method. The questionnaire was only completed for 26.4% of the videos made live. As it was not compulsory, there may have been confounding factors influencing a physician’s decision to complete the questionnaire.

With the exception of the instruction given to the EMDs not to delay the dispatch of an ambulance if they considered the patient’s condition to be serious, there was no standardisation of the use of the EMCC physician, which was left to the discretion of the EMD. The live video facility was only made available to EMCC physicians. As a consequence, it was only employed for second-line patient evaluation. This data is therefore not generalisable to first-line patient evaluations made at an EMCC. It was possible that if first-line responders had also had access to the live video facility, it may have reduced their requirement for a second-line evaluation/EMCC physician input. While the live video facility was made available to all EMCC physicians, the decision to use it was at their own discretion, which, with no guidelines or specific protocols in place, likely resulted in inter-physician variability. Contributing factors may have included their familiarity with the system, previous telemedicine experience and understanding and training regarding suitable indications.

The most frequent indication for its use was to evaluate respiratory function. As respiratory involvement is the main complaint of COVID-19, it likely that this finding was specific to the COVID-19 patients and ungeneralizable to the general patient population calling an EMCC.

It would be interesting to see in future studies how patient management decisions could be influenced if first-line responders also had access to this live video facility. It would also be interesting to further evaluate the time taken to make a patient assessment with or without live video as this is also a consideration when implementing a new system into an EMCC where patient assessments can be time critical and resource limited. This may influence future decisions regarding appropriate patient selection. Finally, it would be interesting to explore further indications which could also benefit from the use of live video in the future.

## 5. Conclusions

During the first two waves of the pandemic, a large number of suspected COVID-19 patients, mainly with influenza-like symptoms or breathing difficulties, were triaged by paramedics, nurses or medical student at the Geneva EMCC. A total of 1 in 5 patients required second-line evaluation by a physician.

Physicians used live video in 22.5% of cases, primarily to assess patients’ breathing and general condition, and found the tool easy to use. The acceptance of live video use by patients or callers was generally very good, but there was a failure rate observed in 22.2% of live video calls.

In the web-based survey, physicians felt that live video influenced their medical decision in 75.7% of assessments. For these live video calls, the most likely outcome was that the patient remained at home without further follow-up, but there were still a significant number of life-threatening emergencies that were caught.

It appears that live video used by physicians integrated into an EMCC contributes to better remote assessment and triage of suspected COVID-19 patients, particularly when these patients present with breathing difficulties. Further studies are needed to confirm these results.

## Figures and Tables

**Figure 1 ijerph-20-03307-f001:**
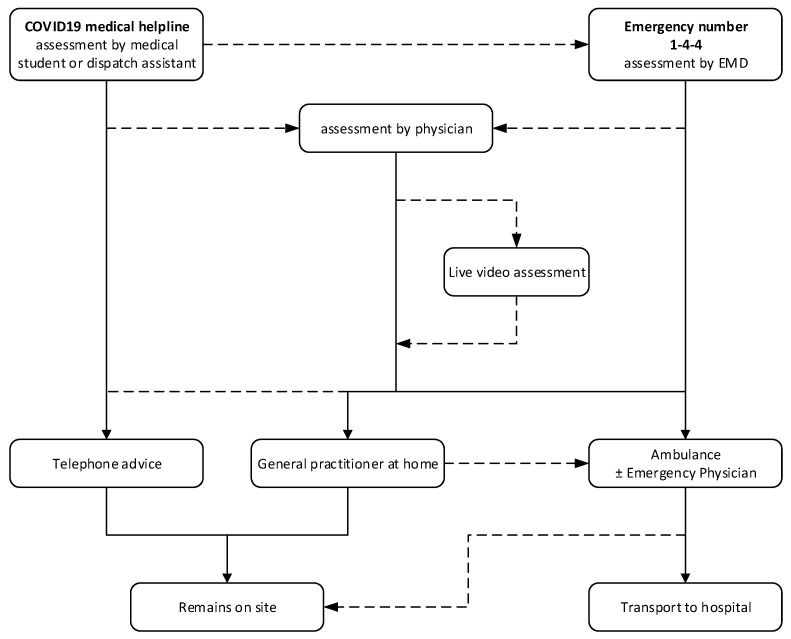
Geneva’s EMCC organization during the COVID-19 pandemic (solid arrows show frequent decisions or routine actions, while dashed arrows show less frequent decisions or possible but more unusual actions).

**Figure 2 ijerph-20-03307-f002:**
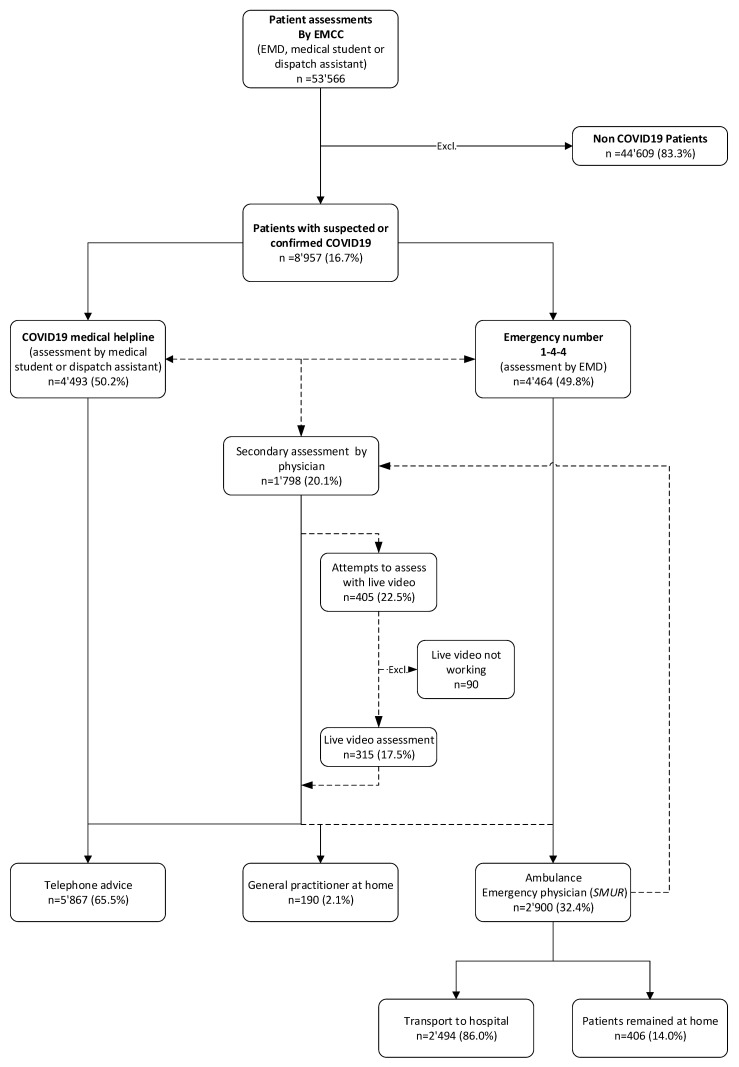
Emergency patient assessment flowchart (1 April 2020–31 April 2021) (solid arrows show frequent decisions or routine actions, whilst dashed arrows show less frequent decisions or possible but more unusual actions).

**Table 1 ijerph-20-03307-t001:** Characteristics of patients with suspected or confirmed COVID-19 infection assessed at Geneva’s EMCC.

	Total	Number	COVID-19 Medical Helpline	*p* Value
	Assessment	1.4.4.		
Patient assessments (n)	8957	4493	4464	
Patient’s sex				
Male, n (% *)	3800 (42.4)	2187 (48.7)	1613 (36.1)	<0.001
Female, n (% *)	4486 (50.1)	2201 (49.0)	2285 (51.2)	
Patient’s age (years, mean ± SD)	49.1 [±21.9]	57.6 [±22.7]	39.8 [±16.6]	<0.001
Level of medical dispatch priority, n (% *)				
Level 1-A (immediate life-threatening emergency, requiring an EP)	305 (3.4)	302 (6.7)	3 (0.1)	<0.001
Level 1-B (life-threatening emergency)	1030 (11.5)	1018 (22.7)	12 (0.3)	<0.001
Level 2 (potential emergency)	1181 (13.2)	1158 (25.8)	23 (0.5)	<0.001
Level 3 (non-urgent situation, but requiring an ambulance in general)	951 (10.6)	848 (18.9)	103 (2.3)	<0.001
Level 4 (non-urgent situation)	5388 (60.2)	1100 (24.5)	4288 (96.1)	<0.001
Unknown	102 (1.1)	67 (1.5)	35 (0.8)	
Main symptom at time of emergency call, n (% *)				
COVID-19 symptoms (rhinorrhoea, throat pain, dry cough, anosmia, agueusia)	4824 (53.9)	779 (17,3)	4045 (90.6)	<0.001
Dyspnoea	2225 (24.8)	2157 (48.0)	68 (1.5)	<0.001
Fever	810 (9.0)	561 (12.5)	249 (5.6)	<0.001
Nausea, vomiting, abdominal pain	129 (1.4)	128 (2.9)	1 (0.0)	<0.001
Other symptoms	969 (10.8)	868 (19.3)	101 (2.3)	<0.001
Medical dispatch decision, n (% *)				
Ambulance ± SMUR	2900 (32.4)	2859 (63.6)	41 (0.9)	<0.001
General Practitioner at home	190 (2.1)	186 (4.1)	4 (0.1)	<0.001
Telephone Advice	5867 (65.5)	1448 (32.2)	4419 (99.0)	<0.001
NACA ** scale in the field, transmitted by paramedics, n (% *)				
>4 (Injuries/diseases with acute threat of life)	86 (1.0)	86 (1.9)	0 (0.0)	<0.001
4 (injuries/diseases which can possibly lead to deterioration of vital signs)	654 (7.3)	649 (14.4)	5 (0.1)	<0.001
3 (injuries/diseases without threat of life but requiring hospital admission)	1447 (16.2)	1432 (31.9)	15 (0.3)	<0.001
<3 (injuries/diseases requiring examination in ambulatory or no examination)	661 (7.4)	645 (14.4)	16 (0.4)	<0.001
Not transmitted (no ambulance)	6109 (68.2)	1681 (37.4)	4428 (99.2)	<0.001
Ambulance dispatched	2900	2859	41	
Ambulance sent but leaves patient on site (no transport), n (% *)	406 (14.0)	391 (13.7)	15 (36.7)	

* All percentages are column percentages. ** NACA: National Advisory Committee for Aeronautic.

**Table 2 ijerph-20-03307-t002:** Live video failures (data from the Instantview^®^ application).

	Failed Call Attempts with Live Video
	90
Reasons for failure of live video related to patients or callers, n (% *)	48 (53.3)
URL link not selected in the SMS	33 (36.7)
Denial of access to the smartphone camera	10 (11.1)
Patient/caller uncomfortable (embarrassed) to be seen on video	5 (5.6)
Reasons for the failure of live video due to technical problems, n (% *)	28 (31.1)
No camera detected on the phone	12 (13.3)
Poor internet connection (Wi-Fi or 4G)	8 (8.9)
SMS message not received on the smartphone	5 (5.6)
Telephone number error	2 (2.2)
IPhone^®^ incompatible	1 (1.1)
Reasons for the failure of live video not determined, n (% *)	14 (15.6)
URL: uniform resource locator	
SMS: short message system	

* All percentages are column percentages.

**Table 3 ijerph-20-03307-t003:** Live video calls assessments by the EMCC Physician (survey).

	Calls with Live Video
	107
Patient/caller acceptance of live video, n (% *)	
Poor acceptance	5 (4.7)
Acceptable or good acceptance	7 (6.5)
Very good or excellent acceptance	95 (88.8)
Ease of use by the physician, n (% *)	
Poor usability	2 (1.9)
Acceptable or good usability	6 (5.6)
Very good or excellent usability	99 (92.5)
Patient’s sex	
Male, n (% *)	41 (38.3)
Female, n (% *)	63 (58.9)
Video call context, n (% *)	
Patient alone at home	28 (26.2)
Patient not alone at home (presence of a close relative)	63 (58.9)
Paramedics on site	16 (15.0)
Indication for the use of live video, n (% *)	
Assessment of breathing	87 (81.3)
Assessment of the patient’s general condition	84 (78.5)
Assessment of patient’s vital signs (other than breathing rate)	17 (15.9)
Other reasons (communication difficulties, difficult psycho-social context)	14 (13.1)
Single indication	37 (34.6)
Multiple indications	70 (65.4)

* All percentages are column percentages.

**Table 4 ijerph-20-03307-t004:** Contribution of the live video call to the outcome of the decision (survey).

	Calls with Live Video
Paramedics not on site, n (% *):	91
Urgent ambulance dispatched (Dispatch Priority Level 1)	7 (7.7)
Non-urgent ambulance dispatched (Dispatch Priority Level 2 or 3)	8 (8.8)
General Practitioner home visit organised	5 (5.5)
Follow-up required (patient to organise)	13 (14.3)
Patient remained at home with advice, no current indication for follow-up	57 (62.6)
Not documented	1 (1.1)
Paramedics on site, n (% *):	16
Re-orientated ambulance to a specific, specialist site	10 (62.5)
Patient remained at home with advice	6 (37.5)
Physician’s evaluation of the live video call contribution to the outcome of the decision, n (% *)	107
Contribution to outcome of the medical decision	81 (75.7)
No Contribution to outcome of the medical decision	1 (0.93)
Insufficient quality of video	17 (15.89)
Not documented	8 (7.48)

* All percentages are column percentages.

## Data Availability

The data presented in this study are available on request from the corresponding author. The data are not publicly available due to incomplete anonymization of the Geneva Emergency Medical Communication Centre remote assessment register (indirect identifiers).

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
