# Peer review of "Contribution of Live Video to Physicians’ Remote Assessment of Suspected COVID-19 Patients in an Emergency Medical Communication Centre: A Retrospective Study and Web-Based Survey"

_ijerph, 2023, doi:10.3390/ijerph20043307_

Round 1

Reviewer 1 Report

I had the privilege of reviewing this manuscript, detailing the use of tele-medicine and, specifically, a video aid on a subset of COVID-19 calls in EMS.

From my point of view, the manuscript seems well-written and sufficiently supported by evidence; the discussion is also robust.

Strengths - thorough description of repurposing and implementation of video calls - ample discussion with comparison with other similar works   Possible Implementations - clarify what criteria prompted deviation of the call to the physician ("difficult situations" is too broad) and what prompted the three final possible ends (telephone advice, GP at home, EMS dispatch); I mean, the authors should specify criteria if they standardized the process, otherwise if not standardized it should be declared among the limits of the study.

Author Response

Thank you very much for this comment.

At the Geneva's emergency medical communication centre, the remote triage process is standardised by applying an adaptation of the Swiss Emergency Triage Scale. After assessing the probability of being in cardiopulmonary arrest, the emergency medical dispatcher has to define a "main symptom" and then define one of the five levels of emergency triage. Only the two least severe levels may lead to a physician being sent home on call. We have explained how triage works at the Geneva Emergency Medical Communication Centre in the article (Symptom-Based Dispatching in an Emergency Medical Communication Centre: Sensitivity, Specificity, and the Area under the ROC Curve) that is quoted in this paper (reference 24). During the pandemic, it was the presence of a physician in the Emergency Medical Communication Centre that was new. The use of this physician was not standardised, and was therefore at the discretion of the emergency medical dispatchers. Emergency medical dispatchers were instructed not to delay the dispatch of an ambulance by making the call to the physician if they felt that the patient's condition was serious. In practice, therefore, calls to the physician were only made for patients who were deemed not to be serious by the emergency medical dispatcher.

We have added the sentence (limitation section) : "With the exception of the instruction given to the EMDs not to delay the dispatch of an ambulance if they considered the patient's condition to be serious, there was no standardisation of the use of the EMCC physician, which was left to the discretion of the EMD."

Reviewer 2 Report

Dear Authors, I read with great interest your work which is very interesting.

I think it deals with a theme that will become in the near future very cogent. I have only minor suggestions that I hope they will improve your paper.

First of all, I would try to shorten it regarding introduction and discussion section. Sometimes for the reader could be difficult to follow the red line.

Second, I would emphasise that during COVID-19 pandemic not only pre-hospital setting has been dramatically modified, but also in-hospital setting. In this regard, please consider this work: doi: 10.23750/abm.v92i2.11159. I believe it adds something about this.

Third: did you required an Ethics Committee evaluation? If not, please add it.

I hope my suggestion will help you.

Best regards

Author Response

Thank you very much for your comments.

First : It is difficult to shorten without "breaking" the structure of the paper. However, we have reconsidered the layout of the paragraphs for more clarity.

Second: Indeed, this aspect of "protecting the system" through better triage and the help of live video has not been mentioned much in our paper. We have added these elements and cited this reference in the manuscript, just before the "limitations" section. The impact was particularly important in relation to hospital emergency departments which were not overloaded, probably because of this. We have added the sentence: "Finally, this advanced remote triage using telemedicine tools may have had a strong impact on avoiding hospital overcrowding during the pandemic, and is therefore one of the tools to be developed for future crises."

Third: We mentioned in the paper (before the acknowledgements section) that the ethics committee did not consider it necessary to evaluate the online survey because the consent of the physicians who responded to the survey was recorded when they logged in to the questionnaire. For the use of retrospective data, the ethics committee had given its agreement.

Lines 415-421:

"Institutional Review Board Statement: Approval for the use of the data from the call assessment registry was given by the Cantonal Commission for Research and Ethics of Geneva (project n°2018-00789) on 12th June 2018.

Informed Consent Statement: The Geneva Cantonal Research and Ethics Commission considered that the patient’s consent could not be obtained and was not necessary for the study. Information about the online survey (purpose and estimated duration) was sent by e-mail to the physicians of the EMCC, and they had to validate their consent electronically before starting the questionnaire."

Reviewer 3 Report

The study is well written with good and professional english, i shall complete it aboout the differences in terms of overcrowding.

I could suggest this article:

Savioli G, Ceresa IF, Gri N, Bavestrello Piccini G, Longhitano Y, Zanza C, Piccioni A, Esposito C, Ricevuti G, Bressan MA. Emergency Department Overcrowding: Understanding the Factors to Find Corresponding Solutions. J Pers Med. 2022 Feb 14;12(2):279. doi: 10.3390/jpm12020279. PMID: 35207769; PMCID: PMC8877301.

for the rrest: simply perfect!!!!

Author Response

Thank you very much for your comments.

Indeed, the aspect of "protecting overcrowding in hospitals" through better triage and the help of live video was not mentioned much in our paper. We have added this element and quoted this reference in the manuscript, just before the "limitations" section. The impact has been particularly important in relation to hospital emergency departments which have not been overloaded, probably because of this.

We have added the sentence: "Finally, this advanced remote triage using telemedicine tools may have had a strong impact on avoiding hospital overcrowding during the pandemic, and is therefore one of the tools to be developed for future crises."